# Differential Analysis of Key Proteins Related to Fibrosis and Inflammation in Soluble Egg Antigen of *Schistosoma mansoni* at Different Infection Times

**DOI:** 10.3390/pathogens12030441

**Published:** 2023-03-11

**Authors:** Ying-Chou Chen, I-An Chen, Shih-Yi Peng, Po-Ching Cheng

**Affiliations:** 1Graduate Institute of Medical Sciences, College of Medicine, Taipei Medical University, Taipei 11031, Taiwan; 2Department of Molecular Parasitology and Tropical Diseases, School of Medicine, College of Medicine, Taipei Medical University, Taipei 11031, Taiwan; b101108099@tmu.edu.tw; 3Drug Metabolism & Pharmacokinetics Department, Institute for Drug Evaluation Platform, Development Center for Biotechnology, Taipei 11571, Taiwan; 4School of Medicine, College of Medicine, Taipei Medical University, Taipei 11031, Taiwan; 5Department of Biochemistry, College of Medicine, Tzu Chi University, Hualien 97004, Taiwan; 6Center for International Tropical Medicine, College of Medicine, Taipei Medical University, Taipei 11031, Taiwan

**Keywords:** soluble egg antigen, *Schistosoma mansoni*, hepatic stellate cells, mass spectrometry, PBMCs, liver fibrosis

## Abstract

Schistosomiasis is a major global health problem. Schistosomes secrete antigens into the host tissue that bind to chemokines or inhibit immune cell receptors, regulating the immune responses to allow schistosome development. However, the detailed mechanism of chronic schistosome infection-induced liver fibrosis, including the relationship between secreted soluble egg antigen (SEA) and hepatic stellate cell (HSC) activation, is still unknown. We used mass spectrometry to identify the SEA protein sequences from different infection weeks. In the 10th and 12th infection weeks, we focused on the SEA components and screened out the special protein components, particularly fibrosis- and inflammation-related protein sequences. Our results have identified heat shock proteins, phosphorylation-associated enzymes, or kinases, such as Sm16, GSTA3, GPCRs, EF1-α, MMP7, and other proteins linked to schistosome-induced liver fibrosis. After sorting, we found many special proteins related to fibrosis and inflammation, but studies proving their association with schistosomiasis infection are limited. Follow-up studies on MICOS, MATE1, 14-3-3 epsilon, and CDCP1 are needed. We treated the LX-2 cells with the SEA from the 8th, 10th, and 12th infection weeks to test HSC activation. In a trans-well cell model in which PBMCs and HSCs were co-cultured, the SEA could significantly induce TGF-β secretion, especially from the 12th week of infection. Our data also showed that TGF-β secreted by PBMC after the SEA treatment activates LX-2 and upregulates hepatic fibrotic markers α-SMA and collagen 1. Based on these results, the CUB domain-containing protein 1 (CDCP1) screened at the 12th infection week could be investigated further. This study clarifies the trend of immune mechanism variation in the different stages of schistosome infection. However, how egg-induced immune response transformation causes liver tissue fibrosis needs to be studied further.

## 1. Introduction

Schistosomiasis is one of the major public health problems worldwide, affecting an estimated 200 million people. Three main types of schistosomiasis affect humans: *Schistosoma mansoni*, *S*. *japonicum*, and *S*. *haematobium*. It is mainly found in East Asian countries, such as Indonesia, the Philippines, and China; sub-Saharan Africa; and central and South American regions, such as the Caribbean island countries and Brazil [1,2,3]. In the lesion produced by the infection, adult female schistosomes lay eggs in the bladder or mesenteric vein, which become entrapped by bladder or hepatic tissues. The secreted egg antigens stimulate T cells to secrete cytokines, causing inflammation and inducing macrophages, lymphocytes, eosinophils, and fibroblasts to surround the eggs layer by layer to form granulomas, which gradually lead to bladder or liver fibrosis [4,5,6]. Clinically, the mechanism of *Schistosoma mansoni* liver fibrosis development is that the female adults lay hundreds of eggs every day, causing hepatic chronic granulomatous inflammation [7,8,9]. The eggs surrounded by granulomas will continuously release and excrete antigens to regulate the surrounding cells, especially hepatic stellate cells (HSCs), thereby inducing liver immunity to Th2 chemotaxis. Granulomatous liver injury then causes the extracellular matrix (ECM) proteins, including collagen, to accumulate excessively and form scar tissue for repair, and form repeated scabs that finally evolve into liver fibrosis [10,11,12]. A key factor in liver fibrosis is the activation of the HSCs, which account for about 5–8% of all liver cells combined [13]. When the immune cells are stimulated by eggs, they secrete TGF-β and bind to the receptor (TGF-β type I receptor, TβRI) on the HSCs, thereby initiating the TGF-β/SMAD signaling pathway. This allows downstream phosphorylation of Samd2, Smad3, and Smad4 to regulate the expression of αSMA in the HSCs to change the cell morphology, transform into myofibroblasts [13,14,15], then activate fibrotic collagen 1, collagen 3, fibronectin, and other ECM synthesis [12,16], eventually leading to liver fibrosis. There is no effective drug to treat schistosomiasis hepatic fibrosis, making it more difficult to understand the pathogenicity and manage the treatment of its diseases.

Previous studies have demonstrated the ability of parasites to form many derivative complexes that can affect host immune responses, limiting the degree of damage and promoting parasite survival [17,18]. Many parasitic infections cause complex acute and chronic inflammatory symptoms in the host, as well as various interactions with parasite or egg antigens [9]. Previous experiments with *S*. *japonicum* infection revealed that TREM-1 expression is closely related to acute inflammatory responses [19]. *S*. *japonicum* SEA can also produce TLR-4 mediated IL-6 and IL-10 cytokines, weakening the INF-γ induced MHC class II gene expression [20]. The host’s response to *S*. *japonicum* infection is to develop alternatively activated macrophages (AAMφ), which cause Th2-dependent chronic inflammation and thus downregulate Th1-induced inflammation to protect the tissues and maintain survival; arginase 1 expression is also closely related to the activation of Th2-type granulomas [21,22]. Chronic schistosome infection increases suppressor T cells and AAMφ [23], which mainly influences Th2 cell growth and decline by regulating the effects of the receptors, including RELM-α and TLR-9, to inhibit the specific granuloma [24,25]. It is also confirmed that chronic schistosomiasis-infected mice can induce low-grade inflammation and TNF-α activation, leading to goblet cell hyperplasia and mucus secretion in the respiratory tract, thereby protecting the host against subsequent pneumovirus infection [26]. However, more research is needed to determine whether this antigen activation has protective effects on the host or causes excessive inflammatory damage to the host.

We previously demonstrated that schistosome eggs can effectively inhibit TLRs, IFN, MHC, and TNF receptor-related gene groups and reduce the degree of subsequent chronic granulomatous inflammation using cDNA microarrays to detect the overall gene transcriptional changes in macrophages after *S*. *japonicum* infection [27]. Furthermore, our most recent findings reveal that in mice infected with *S*. *mansoni*, macrophages differentiate into a significant M2-type AAMφ phenotype after being stimulated by eggs, which subsequently affects lymphocyte differentiation into Th2 and Be2 immunity, resulting in hepatic fibrosis [28]. Schistosomes exude a variety of antigens into host tissue, bind to chemokines, and inhibit immune cell receptors, thereby regulating immune responses to a mode conducive to their development [29,30]. IPSE/alpha-1, the major excreted protein of *S*. *mansoni* eggs, stimulates basophils to secrete IL-4/IL-13, therefore suppressing the inflammatory cytokines secreted by LPS-stimulated monocytes. SEAs play an important role in schistosomiasis-induced host inflammation [31]. We had previously demonstrated that the anti-inflammatory activity of the SEA in an in vitro pneumonia model includes JAK/STAT1 signaling regulation. The SEA has been shown to reduce inflammation in LPS-induced IMR-90 cells by downregulating the IFN-γ-JAK-STAT1 signaling pathway [32]. Zhao et al. demonstrated that the signal transducer and activator of transcription 3 (STAT3) is involved in schistosomiasis-induced liver injury in mice [33]. We also reported that several plant extracts, including *Agaricus blazei*, Casticin, and Schisandrin B, can regulate the transforming growth factor β (TGF-β), which affects the changes of the α-smooth muscle actin (SMA) and collagen 1, consequently altering the severity of the liver fibrosis in infected mice [34,35,36]. However, the detailed mechanism of chronic *Schistosoma* infection-induced liver fibrosis, including the relationship between secreted SEA and HSC activation and how to switch on the regulations of the transmission pathway to affect liver fibrosis, is still quite lacking.

Both schistosomiasis and schistosome eggs have been implicated in the inhibition of allergic airway inflammation induced by OVA or dust mites [37,38,39]. Morais et al. demonstrated that schistosome-derived factors may regulate inflammatory patterns in mice, including acetaminophen-induced liver injury, sodium urate-induced gout, and thioglycolate-induced lung inflammation [40]. However, the mechanism by which schistosome egg antigens regulate inflammatory and fibrotic responses in non-immune cells remains largely unclear. A previous study identified the proteomic analysis of *S*. *japonicum* eggs and egg-derived excreted proteins, both qualitatively and quantitatively. The specialization of the parasite proteome allows for the rapid identification of the selected parasite proteins through the powerful combination of two-dimensional electrophoresis and mass spectrometry, thereby directly providing cDNA into the recombinant protein pathway [41,42]. In terms of *S. mansoni*, Cass et al. used a multidimensional protein identification technique (MudPIT) to characterize the protein files of *S*. *mansoni* egg secretory proteins (ESP) and identified 188 proteins [43]. DeWalick et al. used an in-gel digestion/MS approach to identify at least 45 proteins in the eggshell of *S*. *mansoni* [44]. Carson et al. used shotgun MS to analyze *S*. *mansoni* ESP and found 266 proteins [45]. These findings support the use of mass spectrometry to analyze and screen the anti-inflammatory recombinant protein components, although these studies focus on the ESP or eggshell of the eggs instead of the SEA itself. The direct analysis of tryptic peptides in solution without prior separation for shotgun proteomics could avoid the loss of larger and smaller proteins during multidimensional separation, or the incomplete recovery of proteolytic peptide mixtures following in-gel digestion. In addition, schistosome infection primarily leads to egg production to suppress the TLR and IFN-related gene groups [27], whereas immune cells and chemokines bind to the IFN receptors to increase STAT1-dependent gene expression [46]. Further research is needed to determine whether the complex components of the SEA contain competitive inhibitors. In conclusion, this study explores the complex components of hepatic *S*. *mansoni* SEA at different time points of granulomatous inflammation and fibrotic injury caused by the infection, as well as targets for modulatory action in response to HSC pro-fibrosis. Our results might lead to the development of a novel drug for the treatment of schistosomiasis-induced liver fibrosis.

## 2. Materials and Methods

### 2.1. Parasites and Animals

A Puerto Rican strain of *Schistosoma mansoni* was used in the study. The parasite was maintained and passaged in the intermediate host, Biomphalaria glabrata snails. Male BALB/c mice (aged 6–8 weeks) were purchased from the National Laboratory Animal Center in Taiwan. To obtain parasite eggs for antigen preparation, the mice were percutaneously infected with 120 *S. mansoni* cercariae. The mice were sacrificed at the 8th week (acute stage), 10th week (early fibrosis stage), or 12th week (chronic stage) post-infection. The time points selected were referred to in previous research and our pilot study, which revealed a lot of granuloma formation at the 8th week, which became progressively fibrotic at the 10th week, then led to significant fibrosis at the 12th week [47].

### 2.2. Preparation of SEA

The SEA was prepared based on the methods previously described [32]. After infection, the mice were partially perfused to recover the worms. The livers were dissected and homogenized separately in a 0.85% NaCl solution. The homogenate was passed through a series of sieves with sequentially decreasing pore sizes (420, 177, 105, and 25 μm). The eggs retained in the lowest sieve were collected and resuspended in 0.85% NaCl. The egg suspension was then centrifuged at 3000 rpm for 10 min, and the egg-free supernatant was discarded. The centrifuged eggs were homogenized in phosphate-buffered saline (PBS) and centrifuged at 4 °C and 15,000× *g* for 30 min. The supernatant was sterilized via a 0.22 μm filter. The protein concentration of the SEA was determined by a bicinchoninic acid (BCA) protein assay kit (Pierce, Rockford, IL, USA), which gave the concentrations 0.8, 1.0, and 1.0 mg/mL in the 8th, 10th, and 12th week SEA samples, respectively.

### 2.3. Protein Digestion and LC-MS/MS Analysis

The SEA samples were concentrated down using Amicon^®^ Ultra centrifugal filter units with a molecular weight cut-off of 3 kDa (Merck Millipore) to a final concentration of approximately 20 mg/mL, as determined by the BCA assay kit (Pierce, Rockford, IL, USA). Three technical replicates of the 8th, 10th, and 12th week SEA samples (100 μg) were reconstituted in 50 μL of 50 mM ammonium bicarbonate buffer. Proteins were added with 20 mM dithiothreitol (DTT) to reduce the disulfide bond for 30 min at 50 °C and then alkylated by 50 mM iodoacetamide (IAM) for 60 min in the dark at room temperature. Proteolytic digestion was performed by adding the sequencing grade modified trypsin (Promega, Madison, WI, USA) (trypsin: protein ratio = 1:20, *w*/*w*) and incubating for 12 h at 37 °C with a shaking speed of 1500 rpm. After proteolytic digestion, the trypsin reaction was quenched and acidified with formic acid by lowering the pH value from 2 to 3. The peptides generated by the trypsin digestion were then dried in a vacuum centrifuge and reconstituted in 50 μL of 0.2% acetonitrile/0.1% formic acid. Five microliters of the resulting suspension were analyzed by a TripleTOF 5600 mass spectrometer (AB Sciex, Concord, ON, Canada) in information-dependent acquisition (IDA) mode equipped with an Acquity UPLC I-Class System (Waters). Two elution buffers were used: Buffer A (0.1% formic acid in HPLC-grade water) and Buffer B (0.1% formic acid in 100% acetonitrile). The trypsin-digested peptides (approximately 10 μg) were separated by an Acquity UPLC Peptide BEH C18 Column, 300Å, 1.7 µm, 2.1 mm × 150 mm (Waters) with a linear gradient of 2–10% Buffer B (3 min), 10–40% Buffer B (67 min), and then 40–90% Buffer B (10 min) at a flow rate of 300 nL/min. The full MS spectra of eluted peptides were acquired in positive mode by electrospray over a mass range coverage of 350–1600 (*m/z*), with a charge state of +2 to +5. The product ion spectra were acquired in a data-dependent manner in the TOF-MS scan, exceeding a threshold of 100 counts for the ten most abundant precursor ions selected from the full MS scan with 6 s exclusion after one occurrence. The Analyst TF 1.6.1 software (AB Sciex, Redwood City, CA, USA) was used for data acquisition.

### 2.4. Bioinformatics Analysis

The collected MS spectra were processed using the ProteinPilot™ software v1.5 (Applied Biosystems) against the databases for *S. mansoni* from UniProt (http://www.uniprot.org) accessed on 28 October 2020 (Proteome ID UP000008854, 14174 entries). The false discovery rate (FDR) at both the peptide and protein levels was set at 1%. The proteins identified with at least one unique peptide were considered for further data analysis. Gene Ontology (GO) annotations were assigned based on sequence similarity searches against the GO-annotated proteins in the Swiss-Prot and TrEMBL databases, and literature searches. Protein family (Pfam) analysis was performed by InterPro (https://www.ebi.ac.uk/interpro/) accessed on 1 January 2023. SecretomeP 2.0 accessed on 17 December 2022 (https://services.healthtech.dtu.dk/service.php?SecretomeP-2.0) was used to determine whether the proteins identified were predicted to be secreted through classical or non-classical pathways.

### 2.5. LX-2 Cell Treatment by SEA of Different Infection Weeks

The CCK-8 assay was used to estimate the effects of the SEA on the cytotoxicity of the LX-2 human hepatic stellate cell line (EMD Millipore, San Diego, CA, USA). LX-2 cells (1 × 10^4^/well) were seeded into a 96-well culture plate containing 100 μL DMEM with 2% FBS. After incubation for 24 h at 37 °C in a 5% CO_2_ incubator, the medium was removed and 100 μL of fresh medium containing various concentrations of SEA (0, 2, 4, 6, 8, and 10 μg/mL from the 8th, 10th, or 12th infection weeks) was added into the wells. After incubation for 48 h, the CCK-8 reagent (10 μL) was added to each well and incubated continuously for 1 h. The absorbance value was measured at a wavelength of 450 nm using an M200 Pro microplate reader (Tecan, Männedorf, Switzerland).

### 2.6. Evaluation of Liver Fibrosis Using In Vitro PBMC/LX-2 Co-Culture Model

Cryopreserved human peripheral blood mononuclear cells (PBMCs, Lonza, Walkersville, MD, USA) were resuspended in RPMI-1640 medium containing 10% FBS. The PBMCs were cultured in a 0.4 mm trans-well insert for a 6-well plate at a concentration of 3 × 10^6^ cells per well in a 2 mL culture medium. The cells were incubated for 2 days in the presence of 1 and 2.5 μg/mL of the SEA or the medium alone (unstimulated). The insert was then transferred onto the top of the LX-2 cells (in a 6-well plate, 2 × 10^5^ cells per well) for 2 days. The culture supernatants were collected and stored at −80 °C for TGF-β cytokine measurement. The LX-2 cells were then washed and harvested for real-time PCR analysis.

### 2.7. Enzyme-Linked Immunosorbent Assays (ELISA)

The supernatants of the LX-2 cells were centrifuged at 1000× *g* for 5 min at 4 °C before the ELISA. The levels of TGF-β1 were measured using a commercial ELISA kit (Merck KGaA, Darmstadt, Germany), according to the manufacturer’s protocol. Each sample was evaluated in triplicate.

### 2.8. Real-Time PCR Analysis

The total RNA was extracted from the LX-2 cells using TRIzol reagent (Invitrogen, Carlsbad, CA, USA). The RNA concentration and purity were evaluated by the NanoDrop One (Thermo Scientific, Wilmington, DE, USA). A total RNA of 500 ng was used in the reverse transcription reaction, and cDNA synthesis was performed using a High-Capacity cDNA Reverse Transcription Kit (Applied Biosystems, Foster City, CA, USA), according to the manufacturer’s instructions. For the real-time PCR, 5 μL of cDNA template was added to a 20 μL reaction volume containing each primer and the Power SYBR™ Green PCR Master Mix (Applied Biosystems). The PCR thermocycling parameters were 95 °C for 30 s, followed by 40 cycles of 95 °C for 5 s, and 60 °C for 30 s, performed by the QuantStudio^®^ 5 Real-Time PCR System (Applied Biosystems). All reactions were performed in triplicate. Relative quantification was carried out using the double-delta method (2^−ΔΔCt^). The primer sequences used to amplify the desired cDNA were as follows: Col1A1 forward and reverse primers: 5′-GTCGAGGGCCAAGACGAAG-3′ and 5′-CAGATCACGTCATCGCACAAC-3′; α-SMA forward and reverse primers: 5′-AGGCACCCCTGAACCCCAA-3′ and 5′-CAGCACCGCCTGGATAGCC-3′; and GAPDH forward and reverse primers: 5′-ATGGGGAAGGTGAAGGTCG-3′ and 5′-GGGGTCATTGATGGCAACAATA-3′.

### 2.9. Statistical Analysis

All data are expressed as the mean ± SD. A one-way ANOVA followed by Dunnett’s multiple comparisons test was performed using GraphPad Prism version 6.0 (GraphPad Software, La Jolla, CA, USA). A *p*-value < 0.05 was considered statistically significant.

## 3. Results

### 3.1. Proteomic Analysis of SEA from Different Infected Weeks

We extracted the SEA protein sequences from different weeks using LC–MS/MS analysis. Following database comparisons and literature searches, we identified protein sequences related to fibrosis and inflammation. All the results were screened twice, and only repeated results were accepted after two mass spectrometry analyses to avoid errors. Compared to previous studies that have focused more on egg ESP, we focused primarily on the soluble egg antigen components, which could be easier and faster to prepare for the rapid screening of those fibrosis-associated protein peptides in the cellular evaluation model. To assess the SEA proteome composition at the different stages of *S*. *mansoni*-infected mice, soluble proteins were extracted from the eggs at the 8th, 10th, and 12th infection weeks and characterized by LC–MS/MS. To avoid false positive protein identification, we only kept the proteins identified with at least one unique peptide (a peptide that exists only in one protein of the proteome of interest). We expected to narrow down the targets found in the SEA samples from the different infected weeks and perform a focused literature analysis on these more likely targets. In total, 76, 86, and 57 proteins were identified in the SEA at the 8th, 10th, and 12th infection weeks, respectively (Figure 1A). Of these, only 21 proteins were identified at the 8th week alone, 19 at the 10th week alone, and 7 at the 12th week alone. Thirty-eight proteins of the SEA were common between the three time points post-infection. The SEA from the 8th and 10th infection weeks had 17 proteins in common; the 10th and 12th infection weeks had 12 proteins in common; and the SEA from the 8th and 12th infection weeks did not have any proteins in common. The GO was ascribed to the identified proteins to obtain a basic understanding of the protein functions and predicted locations. The proteins were classified into nine function categories (Figure 1B) and six groups of predicted subcellular location (Figure 1C). The GO analysis revealed that the number of binding-related proteins classified by predicted function at the 8th, 10th, and 12th weeks was relatively large, whereas the proteins classified at 8–12 weeks by subcellular location were mostly located in the cytosol/nucleus, and also many membrane-associated proteins were located in the 12th week. The results of the protein family grouping analysis showed that the heat shock protein featured the most, and that the first three most abundant proteins are the heat shock protein family, the histone family, and the small GTPase family (Figure 1D). Of the 114 identified proteins in the SEA, 58 proteins (50.9% of the total) were predicted by SecretomeP V2.0 to be secreted; these proteins contained either a classic signal peptide leader sequence (13 proteins, 11.4%) or were secreted through a non-classical pathway (45 proteins, 39.5%). (Figure 1E) Concomitantly, 56 proteins (49.1%) were predicted to not be secreted via either pathway. The detailed weekly analysis results are shown in Appendix A.

### 3.2. Proteins Identified in SEA Associated with Hepatic Fibrosis of Schistosomiasis

To further screen out proteins associated with schistosome-induced liver fibrosis, we focused more deeply on the protein subsets at the 10th and 12th infection weeks and excluded proteins at the 8th infection week (Figure 2A). Of the 38 proteins identified in this subset, a total of 11 proteins were inferred to be associated with fibrosis in each cluster based on literature mining (Table 1). These identified proteins were subjected to GO annotation analysis for the protein function (Figure 2B) and subcellular localization (Figure 2C), Pfam analysis for the protein families (Figure 2D), and SecP analysis for the secreted proteins (Figure 2E). After query and analysis, the predicted function included three proteins in the binding group and another three proteins in the enzymatic group; the most being a total of five proteins located in the membrane-associated group, including phospholipid enzymes, glycoproteins, and transmembrane proteins. These proteins were predicted by SecretomeP V2.0 to be five secreted proteins, containing one classic signal peptide and four non-classical secreted proteins. Among them, there are seven proteins in the 10th week and a single protein in the 12th week, and there are also three proteins in both the 10th and 12th week that co-occur. Histones, heat shock proteins, enzymes, and other groups were separated after querying and analysis, and the proteins associated with schistosome-associated hepatic fibrosis were screened out. As shown in Table 1, the proteins have been identified previously confirming their relation to fibrosis and inflammation, such as Sm16, GSTA3, CTP synthase, MMP7, and PI3K in the 10th week group, and GPCRs and EF1-a in the 10th and 12th week groups; which proves that our research is going in the right direction. At the same time, we also found many proteins that are related to fibrosis and inflammation but have not yet been fully documented, including MICOS, MATE1 (the 10th week), 14-3-3 epsilon (the 10th and 12th weeks), CDCP1 (the 12th week), etc. Appendix A shows the detailed analysis results for each classification of the different infection weeks.

### 3.3. Effect of Different Infection Weeks on Activation Induction and Cell Viability of Human LX-2 Hepatic Stellate Cells Induced by SEA

HSC activation is the central process in hepatic fibrosis. In this study, we established a human LX-2 hepatic stellate cell line model to test the effects of the different infection weeks on HSC activation and the viability of the SEA. The eggs at the different infection weeks used to extract the SEA were collected from the livers of the hosts with different degrees of chronic inflammation or fibrosis. The differences in the liver lesions are shown in Appendix A. To determine the influence of the SEA on the activation induction and cell viability of the LX-2 cells, the different concentrations of the SEA from the 8th, 10th, and 12th weeks were incubated with the LX-2 cells (0–10 μg/mL). After 48 h incubation, 2 μg/mL of the SEA from the 8th, 10th, or 12th week of infection did not significantly affect the cell viability. However, it was cytotoxic to the survival of the LX-2 cells at concentrations as high as 4 μg/mL in the SEA of the 8th or 10th week of infection, and as high as 6 μg/mL in the SEA of the 12th week infection (*p* < 0.01, Figure 3). The extent of the cytotoxicity increased in a dose-dependent manner with increasing concentrations of the SEA. Based on the observations, the SEA treatment at concentrations of 1 and 2.5 μg/mL for 48 h was selected for the subsequent experiments.

### 3.4. Effect of SEA on TGF-β Expression in PBMCs Co-Cultured with LX-2

Direct treatment of the LX-2 cells with the SEA induced a significant decrease in mRNA expression in both αSMA and Col1A1 after 48 h incubation. It indicated that the SEA alone could not stimulate the activation of HSCs. The profibrotic effects of the SEA on hepatic fibrosis may be linked to the upregulation of the profibrotic factor TGF-β. To investigate the effects of the SEA on TGF-β production from immune modulatory cells to activate HSCs, each PBMC was pre-stimulated by the SEA population from various post-infection stages before being co-cultured with the LX-2 cells in indirect contact using trans-well inserts (Figure 4A). The PBMCs were seeded in trans-wells with semipermeable membrane inserts with 0.4 μm pores that allowed the free passage of soluble factors but prevented the trans-migration of cells. After PBMC pre-stimulation with 1 μg/mL and 2.5 μg/mL of the SEA for 48 h, the PBMCs were further co-cultured with the LX-2 cells for 2 days. ELISA was used to measure the TGF-β content in the LX-2 culture medium at the end of incubation. The TGF-β expression was increased in the cell groups cultured with SEA pre-stimulation, particularly in the 12th week infection group that significantly and dose-dependently increased more than the control (*p* < 0.05, *p* < 0.01 in 1 μg/mL and 2.5 μg/mL, respectively. Figure 4B).

### 3.5. Effect of SEA on Col1a1 and α-SMA Expression

At the end of the co-culture incubation, RNA was extracted from the LX-2 cells. The Col1a1 and α-SMA mRNA expression levels in the LX-2 cells were used as fibrosis markers. An increased mRNA expression of Col1a1 was observed in the LX-2 cells co-cultured in the presence of the SEA-pretreated PBMCs in comparison to the control LX-2 cells co-cultured PBMCs without SEA treatment (Figure 5A). The 2.5 μg/mL group of the 8th week, the 1 μg/mL group of the 10th week, and the 1 μg/mL group of the 12th week all significantly increased as compared to the control (*p* < 0.05, *p* < 0.0001, and *p* < 0.0001, respectively). The slight increase in mRNA expression of α-SMA was however not significant except in the 1 μg/mL group of the 12th week (*p* < 0.05, Figure 5B).

## 4. Discussion

Fibrosis occurs when connective tissues are overproduced pathologically during tissue repair or the formation of a scar, as in the case of wound healing. Serious fibrosis resulting from inflammatory dysregulation may cause fatal organ failure [62]. The hepatic fibrosis between schistosomes and the infected host is not a one-way reaction but a series of immune network interactions [63]. In each stage of schistosomiasis infection, immune responses with different characteristics will be triggered, and many proteins are involved in the regulation, so the pathogenic mechanism often has multiple effects, resulting in comprehensive pathological symptoms [64,65]. We previously demonstrated that AAMφ induces liver fibrosis to maintain the survival of the eggs by downregulating Th1 responses and egg-induced inflammation during infection [28]. Several studies also proved that the immunosuppressive effect involving AAMφ is indispensable in protecting the host against schistosome-infected lesions, providing tissue repair, and affecting the size and degree of the fibrosis of granulomas [21,65,66,67]. In this study, we first found that egg antigens from different infection weeks contained unique protein compositions, and these unique proteins may have an effect on inducing HSC activation and promoting liver fibrosis. We also found that the fibrotic egg antigen can stimulate PBMC to release more TGF-β, causing HSC activation and granulomatous liver fibrosis in a special PBMC and HSC transmembrane co-culture system.

The immune responses induced by schistosome eggs can be both protective and pathogenic. A previous study used mass spectrometry to qualitatively and quantitatively analyze the protein composition of the *S*. *japonicum* egg secretory proteins (ESP), and the differential expression of proteins by fully mature and immature eggs, isolated from feces and ex vivo adults, and found that mature eggs are more likely to stimulate host immune responses than immature eggs [41]. In our study, we obtained the SEA protein sequences from the different infection weeks through mass spectrometry analysis. After subsequent database comparisons and literature searches, we focused on the SEA components in the 10th and 12th infection weeks and screened out some of the special protein components, especially the fibrosis- and inflammation-related protein sequences. Meanwhile, we found in a PBMC/HSC trans-well co-culture cell model that the SEA containing these protein sequences has the key ability to affect HSC activation, thus enhancing fibrosis or inflammation. The pathogenesis of granulomas is usually mediated through granulomatous fibrosis or necrosis, each of which can occur through multiple mechanisms. Granulomatous macrophages can undergo different types of specialized differentiation, recruit a variety of other cell types, and influence granulomatous structure and function [68]. Proteins from ESP are involved in key functions that lead to the granulomatous inflammatory reaction around the eggs in tissues, but also assist the passage across the gut wall. The study showed 95 of 957 egg-related proteins identified were exclusively found in *S. japonicum* ESP, which are able to stimulate the innate and adaptive immune system through several different pathways. They also revealed that mature eggs with 124 proteins differential to immature eggs are more likely to stimulate host immune responses [41]. Currently, our findings have identified the heat shock protein [51], phosphorylation-associated enzymes or kinases [52,53], Sm16 [54], GSTA3 [55], MMP7 [56], GPCRs [57], EF1-α [58], and other protein components that have been associated with schistosome-induced liver fibrosis. Many studies have confirmed that proteins related to fibrosis and inflammation are closely related to type II immune responses. A key protein, SmKI-1 of *S*. *mansoni*, plays a crucial role in the development of anti-inflammatory diseases by inhibiting the recruitment of neutrophils in the early mouse respiratory tract [40]. In addition, *S*. *mansoni* egg secreted glycoproteins, including Kappa-5, IPSE/alpha-1, and T2 ribonuclease Omega-1, have been identified as carrying immunogenic motifs involved in the major secretion products of *S*. *mansoni* eggs. Kappa-5 can potently bind and blocks DC-SIGN-mediated DC functions [69]. IPSE/alpha-1 in the SEA plays an important role in the regulation of host immunity to drive Th2 responses [31,70]. Omega-1 is a hepatotoxic glycoprotein as is IPSE/alpha-1, which influences the DC function involved in developing Th1 cells [71]. However, due to the focus on screening out the novel components in the SEA at later time points (10th and 12th weeks) for analysis and detection, in order to discuss the components at a single time point in detail; so, compared with previous literature, we found that the number of proteins will be less, and the mainstream egg antigens have also not appeared on our list. In chronically schistosome-infected baboons, *S*. *mansoni* HSP70 can induce early humoral immune responses towards Th2 immunity, whereas *S*. *japonicum* HSP60 can increase the expression of IL-10 and TGF-β to enhance regulatory T cell immunosuppression [53,54]. Similarly, our results have also greatly improved the possibility of analyzing and screening anti-fibrosis and anti-inflammatory immune-related proteins from the SEA components of *S*. *mansoni* eggs, which is worthy of further research and discussion. On the other hand, after sorting, we found many special proteins related to fibrosis and inflammation, but there is insufficient literature to prove that they are related to schistosomiasis infection. It is worthy of our follow-up research, including on MICOS [49], MATE1 [57], 14-3-3 epsilon [58], and CDCP1 [61].

To test the effect of the SEA from the different infection weeks on HSC activation, we treated the LX-2 cells with the SEA from the 8th, 10th, and 12th week of infection. However, it was found that the SEA was cytotoxic to HSCs, and the SEA from the 8th and 10th infection weeks was at a concentration of 4 μg/mL above, while the SEA from the 12th infection week was at a concentration of 6 μg/mL above, inhibiting the survival rates. It has been reported that SEA can inhibit the activation and collagen deposition of HSCs, especially senescent HSCs, which could be related to the survival of the *Schistosoma* eggs in the host [63]. Sjp40 in the SEA of *S*. *japonicum* promotes cellular senescence in HSCs, which shows reduced collagen production and increased degradation [72]. JQ-1 from the SEA relieves liver fibrosis caused by *S*. *japonicum* infections by inhibiting JAK2/STAT3 signaling [73]. It is very important to understand the precise change in the Th1/Th2 immune trend at the different stages of infection for the study of the immune regulation of schistosomiasis [4,74]. The activation of the STAT3 signaling pathway and its mediation of inflammation, oxidative stress, cell proliferation, and apoptosis promote schistosomiasis egg-induced liver injury, which causes subsequent hepatic dysfunction, granuloma formation, and the initiation of fibrosis [33]. However, STAT3 is a double-edged sword in schistosome infections and is involved in the activation and senescence of HSCs, which could be related to different sources of signaling with stimulation [63]. We hypothesize that the SEA activated HSCs mainly by inducing other immune cells in the liver to secrete related cytokines rather than directly activating quiescent HSCs. Therefore, to confirm, we used a trans-well cell model in which the PBMCs and HSCs were co-cultured (Figure 4A), and our results demonstrated that the SEA, particularly the SEA from the 12th infection week, could significantly induce TGF-β secretion. TGF-β plays a crucial part in the development of fibrosis. It has been revealed that HSCs regulated by the TGF-β1/Smad signaling pathway play an important role in liver fibrosis in *S*. *japonicum*-infected mice [75]. TGF-β is secreted by a variety of immune cells through the Smad pathway; further promoting the elevated levels of TIMP1, α-SMA, and collagen 1/2 during the HSC to myofibroblast transition, formatting and remodeling the ECM, and eventually leading to liver fibrosis [76]. Our data also showed that TGF-β secreted by PBMC with the SEA treatment causes activation of the LX-2 cells and leads to the upregulation of hepatic fibrotic markers α-SMA and collagen 1. A recent study reveals that the expression of C-myc and Smad2/3 are upregulated in LPS-treated hepatic stellate cells, while the expression of recombinant thrombospondin 1 and phosphorylation of STAT3 is upregulated in the hepatocytes of mice with liver fibrosis [77]. Based on the above results, the CUB domain-containing protein 1 (CDCP1) screened at the 12th infection week seems to be worthy of further investigation. The CUB domains are mainly found in extracellular proteins and a few plasma membrane-associated glycoproteins. CDCP1 was first described as a transmembrane protein highly expressed in human colorectal and lung tumors [78]. Recent studies suggest that CUB-like domains can bind TGF-β1 and modulate the TGF-β1/Smad signaling pathway by interacting with TGF-β receptors [79,80]. Our results suggest that the CUB domain-containing protein is found in the 12th week of infection. The SEA may enhance TGF-β1 signaling to activate HSC and promote liver fibrosis by expressing α-SMA and collagen 1.

We used mass spectrometry to identify potential chronic inflammation and fibrosis-regulating protein components in *S*. *mansoni* egg antigens and confirmed that the SEA containing these components could promote HSCs activation by inducing TGF-β secretion from PBMCs. In our study, the identified proteins in the different weeks of infection lack important quantitative information about their levels. Therefore, the interpretation of the changes in protein concentration during the infection time is limited. Although this study clarifies the trend of immune mechanism variation in different stages of schistosome infection, how the transformation of the immune response induced by eggs leads to the occurrence of liver tissue fibrosis needs to be explored further. Secondly, although we mainly focused on the soluble egg antigen components in this manuscript, the lack of analysis of the egg secretory antigen (ESP) component is still insufficient to fully explain the course of *Schistosoma* infection-induced liver fibrosis. Therefore, we first completed the analysis and screening of the SEA components at different weeks in our cell model, hoping to serve as a reference for subsequent research on the analysis of the ESP protein components at different infection weeks. Moreover, future studies will analyze these candidate proteins of fibrosis-regulating SEA from different infection time points in-depth, and quickly screen the preliminary anti-fibrosis efficacy of these candidate drugs and their possible intracellular regulatory mechanisms using the hepatic stellate cell assessment model. We anticipate the development of novel worm-derived anti-fibrotic immunomodulatory drug candidates in the future.

## Figures and Tables

**Figure 1 pathogens-12-00441-f001:**
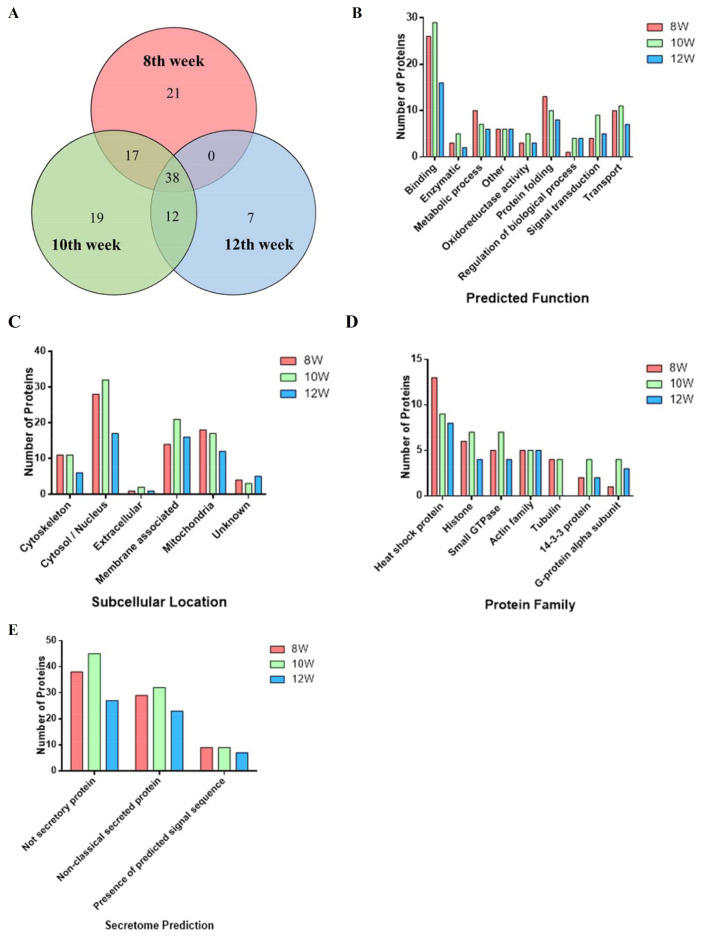
Analysis of the proteins identified between and among the SEA proteins prepared from eggs recovered at the 8th, 10th, and 12th infection weeks. (**A**) Venn diagram; (**B**) predicted function; (**C**) subcellular location; (**D**) protein family (Pfam) analysis of the three most abundant proteins; (**E**) secretome (SecP) analysis.

**Figure 2 pathogens-12-00441-f002:**
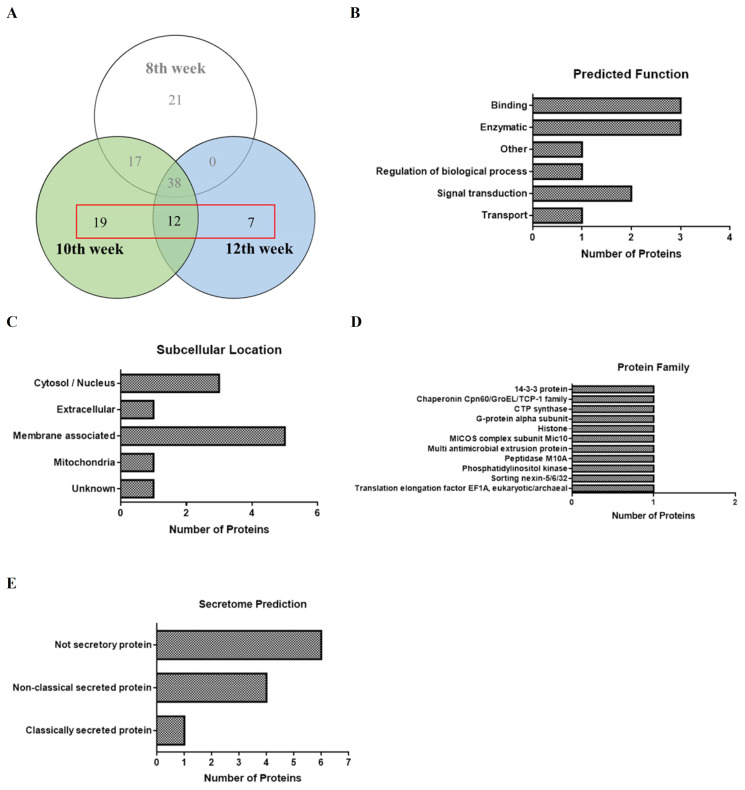
In-depth analysis of the proteins identified between the SEA proteins prepared from the eggs recovered at the 10th and 12th infection weeks. (**A**) Venn diagram of 38 proteins identified in subsets at the 10th and 12th infection weeks. Analysis of 11 proteins speculated to be associated with fibrosis in (**B**) predicted function, (**C**) subcellular location, (**D**) protein family (Pfam) analysis, and (**E**) secretome (SecP) analysis.

**Figure 3 pathogens-12-00441-f003:**
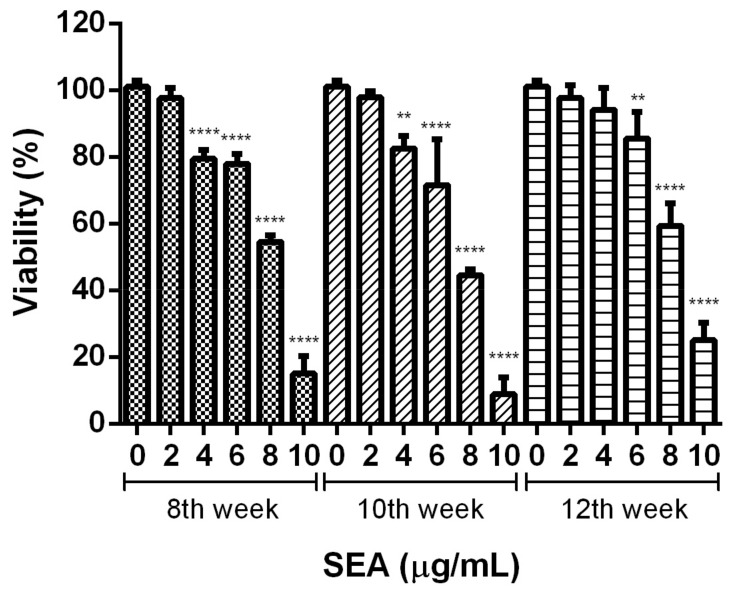
Effect of various concentrations of the SEA on the viability of LX-2 cells. The results of four separate experiments were averaged and shown as a percentage of cell viability compared to the viability of the untreated control. Data are expressed as the mean ± standard deviation (SD). ** *p* < 0.01 and **** *p* < 0.0001 as compared with the untreated control.

**Figure 4 pathogens-12-00441-f004:**
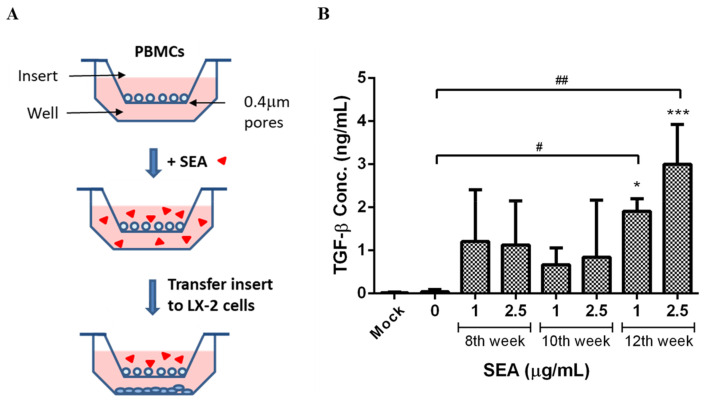
(**A**) Schematic model of the co-culture system of human peripheral blood mononuclear cells (PBMCs) and LX-2 cells. (**B**) PBMCs were pre-stimulated with 1 and 2.5 μg/mL of SEA for 2 days, then co-cultured with LX-2 cells for 2 days. The levels of TGF-β in the LX-2 supernatants were measured by ELISA. The mock group was defined as LX-2 cells without co-culture PBMCs. Data from three separate experiments are expressed as the mean ± standard deviation (SD). * *p* < 0.05 and *** *p* < 0.001 as compared with mock control; ^#^
*p* < 0.05 and ^##^
*p* < 0.01 as compared to the untreated control, represented by the LX-2 cells that had been co-cultured with non-SEA treated PBMCs.

**Figure 5 pathogens-12-00441-f005:**
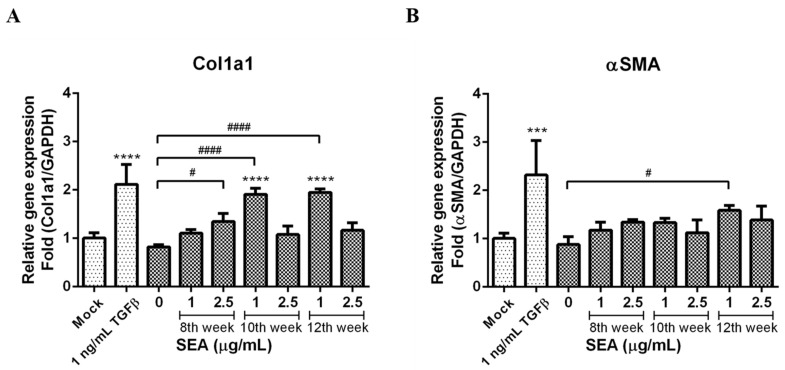
(**A**) Col1a1 and (**B**) α-SMA gene expression in LX-2 cells after co-culture with SEA-stimulated PBMCs, as determined by real-time PCR (Q-PCR) using GAPDH as the reference gene. Data from three separate experiments are expressed as the mean ± standard deviation (SD). *** *p* < 0.001 and **** *p* < 0.0001 as compared with mock control; ^#^
*p* < 0.05 and ^####^
*p* < 0.0001 as compared to untreated control, represented by the LX-2 cells that had been co-cultured without SEA treated PBMCs.

**Table 1 pathogens-12-00441-t001:** Proteins identified from the 10th and 12th week SEA speculated to be associated with liver fibrosis in schistosomiasis.

Infection Weeks of SEA Protein Sequence	Accession Number	Identified Protein	Effect on Fibrotic Pathology	References
Protein sequences that appear only at the 10th week	AF109181.1	Sorting nexin (Sm16)	Sm16 inhibits the activation of macrophages upon invasion, thereby inhibiting the activation of the host adaptive immune response.	[48]
	A0A146MJ03	MICOS complex subunit MIC10	MISCO subunit Mic60 causes mitochondrial dysfunction and cell death, resulting in neurodegeneration, cardiac hypertrophy, and fibrosis.	[49]
	P09792	Glutathione S-transferase class-mu 28 kDa isozyme	GSTA3 is a special target of AKF-PD and is at least partially responsible for its function of reducing ROS accumulation in HSCs and decreasing LPO levels in vivo.	[50]
	A0A5K4F3N0	REVERSED CTP synthase	This enzyme is related to phosphatidylcholine biosynthesis. The ratio of phosphatidylinositol is consistently increased for the more chronic injuries of pulmonary fibrotic diseases.	[51,52]
	A0A3Q0KP46	Matrix metallopeptidase-7 (M10 family)	Host MMPs are necessary for Coronaviruses regarding cell infiltration, survival, and replication, further implicating converging pathways in pulmonary fibrosis.	[53,54]
	A0A3Q0KRX5	Phosphatidylinositol-4,5-bisphosphate 3-kinase	PI3K is related to the overgrowth of human tissues, including visceral fibrous tissue, lipoma overgrowth, and vascular malformations.	[55,56]
	A0A3Q0KMD5	Multidrug and toxin extrusion protein	MATE1 transports TMAO across cell membranes in conjunction with OCT2 and thereby may contribute to its renal secretion. TMAO is associated with kidney fibrosis.	[57]
Protein sequences co-occurring at the 10th and 12th weeks	Q9U491	14-3-3 epsilon	14-3-3 epsilon protein plays a role in signal transduction. Involved in cell proliferation, differentiation, survival, and apoptosis. It is regarded as a potential biomarker of pulmonary fibrosis.	[58]
	G4VAD2	Elongation factor 1-α (EF1-α)	EF1-α acts as a membrane receptor for the cryptic anti-adhesion site of fibronectin and is involved in fibronectin inhibiting cell anchoring and promoting apoptosis or fibrosis.	[59]
	A0A3Q0KCR4	Trimeric G-protein alpha o subunit	It acts as a molecular switch controlling extracellular signals sensed from G protein-coupled receptors (GPCRs) and is associated with fibrosis in humans.	[60]
Protein sequences that appear only at the 12th week	A0A5K4F1G2	CUB domain-containing protein	Also known as CDCP1, is a transmembrane glycoprotein that has been found to function in lung fibroblasts.	[61]

## Data Availability

All data are shown in the main text.

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
