# Peer review of "Differential Analysis of Key Proteins Related to Fibrosis and Inflammation in Soluble Egg Antigen of Schistosoma mansoni at Different Infection Times"

_pathogens, 2023, doi:10.3390/pathogens12030441_

Round 1

Reviewer 1 Report

Title: " Differential analysis of key proteins related to fibrosis and inflammation in soluble egg antigen of Schistosoma mansoni at different infection times"

Introduction: The paper presents a study on the effect of SEA proteins related to fibrosis and inflammation from three different Schistosoma mansoni infection times. The background information provided explains the significance of the research topic and the gap in current knowledge that the study aims to fill. The research question and objectives are clearly stated.

Methods: The study design, sample size, and selection methods must be better described. The data collection procedures should be detailed. It is missing much technical information which will be valuable to replicate the experiments using the same parameters. I will describe some points but this list is not an exhaustive list. The authors must provide all technical information used to produce this dataset.

General remarks:

Double-check all species names and put them in italics;

Describe technical details for protein identification;  

Provide a full list of identified peptides;

Line 153: Please provide SEA concentration by BCA method.

Results: The results are clearly presented, and appropriate statistical tests are used to support the findings. However, I would like to have some key questions addressed before proceeding with the review.

Is this amount of protein compatible with the current literature?

As an example, deWalick et al., 2011 found at least 45 proteins just in the eggshell of Schistosoma mansoni.  In another study, Class et al., 2007 found 188 proteins in the egg secretome. More recently, Carson et al., 2020 found more them 266 proteins in the egg excretory/secretory products. Specifically, there are proteomics studies using a similar approach describing over a thousand proteins that can be detected in SEA, with a broad range of functions on target cells.

How do authors justify these differences and what impact this can bring to the results and discussion? What are the possible limitations of the present study?  

Sometimes authors describe it as post-infection and others p.i. – please define it. 

Overall, this is a well-conducted study that provides valuable insights into SEA proteins related to fibrosis and inflammation. The paper is well-written, and the study presents a significant contribution to the field. However key questions must be addressed before proceeding to further publication steps.

Author Response

Reviewer1:

Comments and Suggestions for Authors

Title: " Differential analysis of key proteins related to fibrosis and inflammation in soluble egg antigen of Schistosoma mansoni at different infection times"

Introduction: The paper presents a study on the effect of SEA proteins related to fibrosis and inflammation from three different Schistosoma mansoni infection times. The background information provided explains the significance of the research topic and the gap in current knowledge that the study aims to fill. The research question and objectives are clearly stated.

Ans: Thanks for the reviewer's support, we hope this manuscript could help to analyze and research about liver fibrosis caused by the components of schistosome egg antigens.  

Methods: The study design, sample size, and selection methods must be better described. The data collection procedures should be detailed. It is missing much technical information which will be valuable to replicate the experiments using the same parameters. I will describe some points but this list is not an exhaustive list. The authors must provide all technical information used to produce this dataset.

General remarks:

  1. Double-check all species names and put them in italics;

Ans: Thanks for the reviewer's comment. We have checked and corrected all species names in italics. (All species names in the text)

  1. Describe technical details for protein identification;  

Ans: Thanks for the reviewer's suggestion. We have revised the Methods sections of 2.3. (Protein digestion and LC-MS/MS analysis) and of 2.4. (Bioinformatics analysis) to provide technical details for protein identification. (page4, line169-173, 180-181, 190-194; line 199-204)

  1. Provide a full list of identified peptides;

Ans: Thanks for the reviewer's comment. The full list of identified proteins has been attached in the supplemental Table S1. (page14, line542-546, Supplemental S1)

  1. Line 153: Please provide SEA concentration by BCA method.

 Ans: Thanks for the reviewer's comment. We have added the SEA concentrations determined by BCA assay in Methods section of 2.2. (page4, line166-167)

Results: The results are clearly presented, and appropriate statistical tests are used to support the findings. However, I would like to have some key questions addressed before proceeding with the review.

  1. Is this amount of protein compatible with the current literature?

As an example, deWalick et al., 2011 found at least 45 proteins just in the eggshell of Schistosoma mansoni.  In another study, Class et al., 2007 found 188 proteins in the egg secretome. More recently, Carson et al., 2020 found more them 266 proteins in the egg excretory/secretory products. Specifically, there are proteomics studies using a similar approach describing over a thousand proteins that can be detected in SEA, with a broad range of functions on target cells.

Ans: We thank the reviewers for their suggestions, and we have added these relevant references in the Introduction and Discussion sections for comparison with our data, which we hope will contribute to the acceptance of this manuscript. Compared to previous studies that have focused more on egg ESP, we focused primarily on soluble egg antigen components, which are easier and faster to prepare for rapid screening of those fibrosis-associated protein peptides in the cellular evaluation model of this manuscript. Second, we set to use the UniProt S. mansoni Swiss-Prot protein database to exclude potentially repeated or similar sequences, or proteins that appeared only once. In our study, we used the shotgun MS approach: the concentrated SEAs were reduced, alkylated and proteolytic digested, and the peptide mixtures were analyzed directly in solution without gel-separation. The collected MS spectra of the 8th, 10th and 12th week SEA were processed using ProteinPilot™ software (Applied Biosystems) against the database for S. mansoni (downloaded from UniProt http://www.uniprot.org ; Proteome ID UP000008854 ; 14174 protein numbers). Peptides were considered identified if they passed the scoring thresholds (the false discovery rate (FDR) at both the peptide and protein levels was < 1%). There were 129, 154 and 139 proteins found in the 8th, 10th and 12th weeks, respectively. To avoid false positive protein identification, we only kept the proteins identified with at least one unique peptide (a peptide that exists only in one protein of the proteome of interest). After removing the proteins without unique peptide searched, the number of proteins identified in the 8th, 10th and 12th weeks were down to 76, 86 and 57, respectively. Therefore, we thought the numbers of proteins identified in our study is compatible with the current literature (injected protein amount per sample: Carson et al., 2020 vs. our study = 15 ug vs. 10 ug). We expected to narrow down the targets found in the SEA samples from different weeks and perform a focused literature analysis on these more likely targets. Therefore, we focus on screening out the novel components in SEA with later stages of time-point for analysis and detection, instead to discuss the components at a single time point in detail. We also revised the description of Result 3.1 section to explain this result. (page3, line126-136; page4, line169-173, 180-181, 190-194; line 199-204; page5, line263-265, 268-272; page12, line445-448; Ref. 41, 43-45)

  1. How do authors justify these differences and what impact this can bring to the results and discussion? What are the possible limitations of the present study?  

Ans: Thanks for the reviewer's comments. As mentioned above, we have provided relevant descriptions and added references to explain these differences in the Results and Discussion sections, and we have also added possible limitations of our study in the Discussion section. We hope these revisions will help improve the previous deficiencies and make this manuscript more understanding. (page5, line263-265, 268-272; page12, line445-448; page14, line524-526, 529-538)

  1. Sometimes authors describe it as post-infection and others p.i. – please define it. 

Ans: We are sorry for the confusing description in inconsistently using post-infection and p.i.  In order to correct the mistake, we uniformly use “the 8th, 10th, and 12th infection weeks” to replace “8, 10 and 12 weeks post-infection” and “8, 10 and 12 weeks p.i.” throughout the manuscript. We hope these revision could make the description clearer. (Throughout the text)

Overall, this is a well-conducted study that provides valuable insights into SEA proteins related to fibrosis and inflammation. The paper is well-written, and the study presents a significant contribution to the field. However key questions must be addressed before proceeding to further publication steps.

Ans: Again, we are very appreciate with the reviewer's suggestion and support, we hope this revision can lead to this manuscript to be publish.  

Reviewer 2 Report

Not use yellow in the graphs 

Author Response

Reviewer2:

  1. Not use yellow in the graphs

Ans: Thanks for the reviewer's comment. We have revised the yellow color used in Fig 1A, Fig 2B-E, and Fig 4A to follow reviewer’s comment. (Fig1, 2 and 4)

Reviewer 3 Report

The manuscript ID-2215824 was investigating the major proteins from soluble egg antigen (SEA) preparing from Schistosoma mansoni eggs.  The parasite eggs were collected from mouse liver tissues after S. mansoni infections for 8, 10 and 12 weeks.  The mass spectrophotometry along with several bioinformatic analysis including ProteinPilotTM software were used for secreted protein predictation.  The authors treated LX-2 cells with SEA to optimize the non-toxic condition for later PBMC/LX-2 co-cultured in vitro model to evaluate liver fibrosis after SEA treatment.  The secreted TGF-beta from LX-2 cells after treating with 2.5 ug/ml of SEA of 12 weeks post parasite infection shown highest amount comparing with 10 and 8 weeks.  The cell transcript expression of Col1a1 and alpha-SMA were up-reguation after treatment with SEA from 10 weeks or longer S. mansoni infection.  The report in this study about liver fibrosis related- and/or induced- protein from parasite eggs would be pentential markers to futher drug developments and related topics. 

In my opinion, this manuscript has a great objective and outcome for the scientific field, but the current version has not reached the satisfaction to be accepted for publication yet.  The manuscirpt would be suitable for publications after authors clarify and/or discuss few major concerns as below.

-        Because of the spectrum of mixed immature and mature eggs from mouse livers at 8-, 10- and 12-weeks including number of eggs from those post infection weeks are expected to be different.  Each time point of egg collection, some of those eggs would be activated by mouse immune response in vivo, and some active protein(s) might be secreted in the mouse model.  To collect the eggs and proceed for SEA may not the assist for fibrosis model in vitro because lack of excretory-secretory protein (ESP) in vitro study.  Especially, authors have mentioned about criteria for bioinformatic to search both classic signal peptide and non-classic signal peptide.  Please discuss and add reference(s) if needed.

-        It is unclear how authors excluded the major antigenic egg proteins such as kappa-5, interleukin-4-inducing principle (IPSE) and omega-1.  Please provide more information and/or discuss how these proteins could be excluded to be involve in parasite egg induced-liver fibrosis.

-        Eventhrough, author used PBMC/LX-2 co-culturing system to mimic liver fibrosis, but the lack of information of mouse liver fibrosis at each time of egg collections.  Did authors proceed any immunohistochemistry study of liver tissues?  When was liver fibrosis in mouse model started?  Was it 8 weeks or later?  Please clarify in material and methods and/or discussion.

Author Response

Reviewer3:

The manuscript ID-2215824 was investigating the major proteins from soluble egg antigen (SEA) preparing from Schistosoma mansoni eggs.  The parasite eggs were collected from mouse liver tissues after S. mansoni infections for 8, 10 and 12 weeks.  The mass spectrophotometry along with several bioinformatic analysis including ProteinPilotTM software were used for secreted protein predictation.  The authors treated LX-2 cells with SEA to optimize the non-toxic condition for later PBMC/LX-2 co-cultured in vitro model to evaluate liver fibrosis after SEA treatment.  The secreted TGF-beta from LX-2 cells after treating with 2.5 ug/ml of SEA of 12 weeks post parasite infection shown highest amount comparing with 10 and 8 weeks.  The cell transcript expression of Col1a1 and alpha-SMA were up-reguation after treatment with SEA from 10 weeks or longer S. mansoni infection.  The report in this study about liver fibrosis related- and/or induced- protein from parasite eggs would be pentential markers to futher drug developments and related topics.

In my opinion, this manuscript has a great objective and outcome for the scientific field, but the current version has not reached the satisfaction to be accepted for publication yet.  The manuscirpt would be suitable for publications after authors clarify and/or discuss few major concerns as below.

Ans: Thanks for the reviewer's support, we hope this manuscript could help to analyze and research about liver fibrosis caused by the components of schistosome egg antigens.  

  1. Because of the spectrum of mixed immature and mature eggs from mouse livers at 8-, 10- and 12-weeks including number of eggs from those post infection weeks are expected to be different. Each time point of egg collection, some of those eggs would be activated by mouse immune response in vivo, and some active protein(s) might be secreted in the mouse model.  To collect the eggs and proceed for SEA may not the assist for fibrosis model in vitro because lack of excretory-secretory protein (ESP) in vitro study.  Especially, authors have mentioned about criteria for bioinformatic to search both classic signal peptide and non-classic signal peptide.  Please discuss and add reference(s) if needed.

Ans: We are much appreciated reviewer's comments. In this manuscript, we focus primarily on soluble egg antigen components, which are easier and faster to prepare for rapid screening of those fibrosis-associated protein peptides in our cellular evaluation model. Furthermore, given the previous Proteomics research has mostly focused on the analysis of the secretory antigen (ESP) components of eggs, which is relatively difficult to obtain from materials with different weeks of liver fibrosis; especially the survival rate of eggs in the chronic stage is relatively low. So we first complete the SEA component analysis and screening detection of different weeks in our cell model, as a reference for subsequent research on the ESP protein component analysis of different infection weeks. Second, we focus on screening out the novel components in SEA with later stages of time-points (10th and 12th weeks) for analysis and detection, instead to discuss the components at a single time point in detail; so compared with previous literature, we found that the number of proteins will be relatively less, and these mainstream egg antigens have also not appeared in our list. In addition, to avoid false positive protein identification, we only kept the proteins identified with at least one unique peptide (a peptide that exists only in one protein of the proteome of interest). Therefore, we added related references and revised some description in Discussion sections to explain these problems and avoid confusing. (page3, line126-136; page4, line169-173, 180-181, 190-194; line 199-204; page5, line263-265, 268-272; page12-13, line445-448, 456-468; Ref. 41, 43-45)

  1. It is unclear how authors excluded the major antigenic egg proteins such as kappa-5, interleukin-4-inducing principle (IPSE) and omega-1. Please provide more information and/or discuss how these proteins could be excluded to be involve in parasite egg induced-liver fibrosis.

Ans: Thanks for the reviewer's comment. We have supplemented references and description of the egg immunogenic proteins Kappa-5, IPSE and Omega-1 involved in the host-parasite interactions in the Discussion section. In this study, we focused on the SEA components found in the 10th and 12th weeks and more in-depth discussion in new fibrosis- and chronic inflammation-related proteins based on database analysis and literature mining. So compared with previous literature about ESP, these mainstream egg antigens have also not appeared in our list. (page5, line263-265, 268-272; page12, line445-448; Ref. 31, 69-72)

  1. Eventhrough, author used PBMC/LX-2 co-culturing system to mimic liver fibrosis, but the lack of information of mouse liver fibrosis at each time of egg collections. Did authors proceed any immunohistochemistry study of liver tissues?  When was liver fibrosis in mouse model started?  Was it 8 weeks or later?  Please clarify in material and methods and/or discussion.

Ans: Thanks to the reviewers for their comments. We provide information on liver fibrosis in mice at different time points in Supplementary Information Figure 3. We also added a literature as a reference for liver pathological processes, and revised the related description in Methods section of 2.1 and Results section of 3.3. We hope these revision can help to explain possible problems. (page3, line152-155, page9, line342-345; page14, line548-562; Ref. 47; Supplemental S3)

Round 2

Reviewer 1 Report

The authors have provided an improved version of the manuscript, which I appreciate. However, I would like to request access to the full list of identified peptides, including the raw peptide sequence. This is a crucial step in protein identification, as noted by the authors, and I believe that making this file fully available would be highly beneficial for future research in this field. 

Author Response

Ans: Thanks for the reviewer's suggestion, we have added the total protein list be identified as below; and supply the full list file with raw peptide sequence in supplemental information (please see the attachment). We hope this revision could help this manuscript to be accepted. Thank you again for the comment and support from reviewer.

Reviewer 3 Report

The authors have clarify my questions and comments.  The authors also have included the relevant information to clarify the issues in the discussion which will be useful for the reader and cover the non-clarifications.

I think, the revised version is suitable to be published.

Author Response

Ans: Thank you again for the comment and support from reviewer.